# COVID-19 Vaccine Hesitancy and Its Associated Factors in Japan

**DOI:** 10.3390/vaccines9060662

**Published:** 2021-06-17

**Authors:** Ryo Okubo, Takashi Yoshioka, Satoko Ohfuji, Takahiro Matsuo, Takahiro Tabuchi

**Affiliations:** 1Department of Clinical Epidemiology, Translational Medical Center, National Center of Neurology and Psychiatry, Tokyo 187-8551, Japan; 2Center for Innovative Research for Communities and Clinical Excellence (CiRC2LE), Fukushima Medical University, Fukushima 960-1295, Japan; yoshioka.takashi.52a@kyoto-u.jp; 3Department of Public Health, Graduate School of Medicine, Osaka City University, Osaka 545-8585, Japan; satop@med.osaka-cu.ac.jp; 4Division of Infectious Diseases, St. Luke’s International Hospital, Tokyo 104-8560, Japan; takahirom1226@gmail.com; 5Cancer Control Center, Osaka International Cancer Institute, Osaka 541-8567, Japan; tabuchitak@gmail.com

**Keywords:** COVID-19, vaccine, vaccine hesitancy, Japan, longitudinal study, psychological distress, social determinants of health, socioeconomic status

## Abstract

The vaccine confidence index in Japan is one of the lowest worldwide. This study aimed to examine the proportion of COVID-19 vaccine hesitancy in the Japanese population using a larger sample and more robust statistical methods than previously, and to identify factors associated with vaccine hesitancy. We conducted a nationwide, cross-sectional Internet survey on 8–26 February 2021, and calculated the proportion and odds ratios for vaccine hesitancy. Among 23,142 responses analyzed, the proportion of COVID-19 vaccine hesitancy was 11.3% (10.9–11.7%). The proportion was higher among younger respondents and female respondents, and especially among younger female respondents (15.6%) compared with the lowest proportion among older male respondents (4.8%). The most cited reason for not getting vaccinated was concerns about adverse reactions in more than 70% of the respondents. The proportion of COVID-19 vaccine hesitancy in Japan was comparable to that in previous studies overseas, and the proportion among younger respondents was more than double that among older respondents. Factors associated with the hesitancy were female sex, living alone, low socioeconomic status, and presence of severe psychological distress, especially among older respondents. Thus, adequate measures should be taken to ensure that vaccines are delivered to people with these factors.

## 1. Introduction

Coronavirus disease 2019 (COVID-19) has now spread worldwide. To date, it has infected more than 174 million people, of whom 3.7 million have died, and these numbers continue to rise [1]. In addition to protective measures such as social distancing and quarantine, establishing herd immunity through vaccines will be the best strategy for ending the COVID-19 pandemic.

Vaccine hesitancy is an obstacle to the establishment of herd immunity and is defined as delayed acceptance, reluctance, or refusal of vaccination even though there are available vaccination services [2,3]. In fact, it has been identified as one of the top 10 threats to global health in 2019 by the World Health Organization (WHO) [3]. Vaccine hesitancy is caused by a complex decision-making process that is affected by a variety of contexts, such as individual, group, and vaccine-specific factors including media, history, religion, culture, and socioeconomics [2].

In a recent systematic review of 28 nationally representative samples that consisted of 58,656 participants from 13 countries, the proportion of COVID-19 vaccine hesitancy was reported to be 20% (13–29%) [4]. However, no studies from Japan were included in the review. To date, two studies have examined vaccine hesitancy in Japan, but the reported proportions of the hesitancy have shown very large fluctuations. In the September 2020 survey (*N* = 1100) [5], the proportion was 12.3%, while in the January 2021 survey (*N* = 3000) [6], it was 37.9%. The vaccine confidence index in Japan has been one of the lowest worldwide [7]; this is possibly because the nation’s Ministry of Health, Labor and Welfare suspended proactive recommendation of the human papillomavirus vaccine in 2013 as a result of general public concern about the safety of the vaccine [8,9,10]. Therefore, there is concern about domestic acceptance of the COVID-19 vaccine in Japan, and examination of vaccine hesitancy is needed.

Furthermore, studies have consistently reported that COVID-19 vaccine hesitancy is possibly associated with female sex, young age, low income, and low education level [4]. However, both of the abovementioned studies in Japan examined factors that influence willingness to get vaccinated against COVID-19, whereas risk factors for vaccine hesitancy have remained unclear. Vaccine willingness and hesitancy are not just two sides of the same coin. In addition to “willingness” and “hesitancy”, there is also the attitude of being “unsure” about vaccines. Therefore, it is necessary to distinguish between willingness and hesitancy, and to examine the risk factors of COVID-19 vaccine hesitancy among the Japanese population.

Therefore, in this study, we aimed to examine the proportion of COVID-19 vaccine hesitancy among the Japanese population using a larger sample than employed in previous studies, and to identify factors associated with COVID-19 vaccine hesitancy.

## 2. Materials and Methods

### 2.1. Study Design and Participants

This was a cross-sectional study that used data collected by our nationally representative survey called the Japan “COVID-19 and Society” Internet Survey (JACSIS). Details of the JACSIS are described in our previous studies [11,12]. Briefly, we conducted a baseline survey during September 2020 and collected data on 28,000 male and female respondents aged 15–79 years throughout Japan selected from 224,389 panelists registered with a large Internet survey agency [13]. Selection was done by random sampling stratified by sex, age, and prefecture (a large area with a local government that falls directly under the national government and that forms the first level of jurisdiction and administrative division in Japan). All of Japan’s 47 prefectures were represented in the survey. Participants provided web-based informed consent before responding to the online questionnaire, and were allowed to stop participation in the survey at any point.

After the baseline survey, we conducted a follow-up survey from 8 to 26 February 2021. Of the 28,000 participants in the baseline survey, 24,059 participated in the follow-up survey. In addition to the participants, we recruited 1941 new general residents aged 15–79 years without stratification by age, sex, or prefecture, for a total of 26,000 study participants. COVID-19 vaccine hesitancy was assessed only in the follow-up survey.

All procedures described in this work comply with the ethical standards of the relevant national and institutional committees on human experimentation and with the Helsinki Declaration of 1975 and its 2008 revision. The Research Ethics Committee of the Osaka International Cancer Institute reviewed and approved the study protocol (approved 19 June 2020; approval no. 20084). The Act on the Protection of Personal Information in Japan was followed by the Internet survey agency. As an incentive for study participation, the participants received credit points called “Epoints”, which can be used for online shopping and cash conversion. Although the exact value of each Epoint was not disclosed at the Internet survey agency’s request, 1 Epoint was assumed to be equivalent to around 100 yen (about 1 US dollar).

### 2.2. Assessment of COVID-19 Vaccine Hesitancy

The survey asked the participants, “What do you think about vaccination against new coronavirus infections?” The participants answered by selecting one of the following three options: “I want to be vaccinated”, “I want to be vaccinated after seeing how it goes”, and “I don’t want to be vaccinated”. Those who answered “I want to be vaccinated” and “I want to be vaccinated after seeing how it goes” were defined as “Intend”, while those who answered “I don’t want to be vaccinated” were defined as “Hesitant”.

The survey also asked the Intend group their reasons for getting vaccinated against COVID-19 and the Hesitant group their reasons for not getting vaccinated. Multiple answers were allowed, so the sum of the proportions for each reason did not necessarily add up to 100%.

### 2.3. Assessment of Possible Factors Related to COVID-19 Vaccine Hesitancy

Demographic factors found to be possibly related to COVID-19 vaccine hesitancy were as follows: sex, age group (younger (15–39), middle-aged (40–64), or older (65–79)), income level, marital status (married, never married, or widowed/divorced), living alone (yes or no), occupation (essential worker in the food industry, health care professional, other occupation, or unemployed), highest level of education, smoking, drinking, major comorbidities, and worsening of employment situation due to COVID-19 based on previous studies [6,14]. We referred to the National Health and Nutrition Examination Survey, which was conducted with a population-based nationally representative cohort in the United States [15]. In that survey, younger, middle, and older age groups were defined as 20–39, 40–64, and ≥65 years, respectively. Those working in the agriculture, forestry, fisheries, and food service industries were defined as essential workers in the food industry [16]. Major comorbidities were categorized into hypertension, diabetes mellitus, asthma or chronic obstructive pulmonary disease (COPD), cardiovascular disease, cerebrovascular disease, cancer, chronic pain, and psychiatric disorder [17].

Other possible factors related to COVID-19 vaccine hesitancy included personal history of COVID-19 infection (yes or no), fear of COVID-19-induced death (yes or no), perceived likelihood of getting infected with COVID-19 themselves (yes or no), distrust toward the government (yes or no), distrust toward government policy on COVID-19 (yes or no), the thought of embarrassment of getting infected with COVID-19 themselves (yes or no), severe psychological distress, and living in a prefecture with a high proportion of COVID-19 cases.

Severe psychological distress was assessed by using the Kessler Psychological Distress Scale (K6) [18]. This 6-item scale measures psychological distress in general populations and is used in many epidemiological studies. Participants who score 13 or higher are defined as having severe psychological distress [18,19]. The proportion of COVID-19 cases in a prefecture was calculated based on the number of cases of COVID-19 infection in the prefecture between 15 January 2020 and 7 February 2021 [20]. We divided the number of cases in each prefecture by that prefecture’s population as of 1 October 2019. The average number of COVID-19 cases per million population in Japan through 7 February 2021 was 3203. The participants were divided into two groups according to whether the number of cases in the prefecture where they lived was more or less than the average (Appendix A).

### 2.4. Statistical Methods

There were differences in sociodemographic status between the Internet survey participants in the present study and the general Japanese population, so we attempted to make adjustments for all analyses. We adjusted for differences between the Internet survey respondents and the general population (e.g., younger individuals are more likely to participate in Internet surveys compared with older individuals) by applying an inverse probability weighting (IPW) method using propensity scores. We calculated the propensity scores by logistic regression analysis and used sex, age, and socioeconomic factors to adjust for differences between the Internet survey respondents and respondents of the 2016 Comprehensive Survey of Living Conditions of People on Health and Welfare (CSLCPHW), which is a widely used sample that is population-based and representative of the Japanese population (see the Appendix A for details).

The abovementioned variables and reasons for getting vaccinated against COVID-19 or not getting vaccinated were calculated as proportions using the abovementioned IPW stratified by responses regarding vaccine hesitancy [21,22]. We also calculated proportions of COVID-19 vaccine hesitancy stratified by age and sex. To identify factors associated with COVID-19 vaccine hesitancy, we calculated odds ratios (ORs) and 95% confidence intervals (CIs) for the hesitancy using a logistic regression model with IPW based on propensity score. In these analyses, we entered all the abovementioned variables as independent variables into the models. A two-tailed *p*-value of less than 0.05 was considered statistically significant. All analyses were performed using SAS software version 9.4 (SAS Institute, Cary, NC, USA).

## 3. Results

From among the 26,000 total participants, we excluded 2858 participants whose responses raised concerns about whether they were answering consciously (some participants may have been answering without reading the questions in order to complete the survey quickly) (see the Appendix A for details). Thus, data from 23,142 (89.0%) survey participants were included in the analyses, and their baseline characteristics are shown in Table 1. Among the participants, 49.2% were female and 25.5% were aged 65–79 years. Demographic data before weighting are shown in Appendix A.

As shown in Table 2, the overall proportion of vaccine hesitancy was 11.3% (10.9–11.7%). The proportion was higher among younger respondents and female respondents, and especially among young female respondents (15.6%) compared with the lowest proportion among older male respondents (4.8%). The proportion of those who expressed an intention of getting vaccinated against COVID-19 according to the three abovementioned response options are shown in Appendix A.

As shown in Table 3, among respondents who did not want to get vaccinated against COVID-19, the top reason given was concern about adverse reactions, which was mentioned by more than 70% of the Hesitant group, followed by doubts about the efficacy of the vaccine, which was mentioned by 20%. Each of the following three reasons for getting vaccinated were given by about 30% of the participants in the Intend group: “I worry about getting infected with COVID-19”, “I don’t want to infect my family or other people around me”, and “I think it is necessary for society to be vaccinated”. Reasons for getting vaccinated against COVID-19 or not getting vaccinated, stratified by age are shown in Appendix A.

As shown in Table 4, factors associated with COVID-19 vaccine hesitancy were female sex, younger age, low income, married, living alone, occupation outside the health care profession or outside essential work in the food industry, low educational level, current alcohol use, presence of comorbidities (diabetes mellitus, psychiatric disorder), fear of COVID-19-induced death, distrust toward the government, distrust toward government policy on COVID-19, and presence of severe psychological distress. In testing for multicollinearity, the variance inflation factor ranged from 1.02 to 2.01, indicating that there was no multicollinearity in this multiple regression analysis. Appendix A also shows the odds ratio of the COVID-19 vaccine hesitancy stratified age group. Factors significantly associated with COVID-19 vaccine hesitancy among the younger (20–39 years) respondents were as follows: low income, married, living alone, current alcohol use, presence of comorbidities (hypertension, diabetes mellitus, asthma or COPD, chronic pain), fear of COVID-19-induced death, distrust toward the government, and presence of severe psychological distress. Among the middle-aged (40–64) respondents, the significant factors were as follows: low income, married, living alone, occupation outside the health care profession or outside essential work in the food industry, current alcohol use, presence of comorbidities (diabetes mellitus, chronic pain, psychiatric disorder), fear of COVID-19-induced death, distrust toward government policy on COVID-19, and presence of severe psychological distress. Among the older (65–79 years) respondents, the significant factors were as follows: female sex, low income, married, living alone, occupation outside the health care profession or outside essential work in the food industry, low educational level, current alcohol use, presence of comorbidities (diabetes mellitus, cardiovascular disease, cancer, chronic pain, psychiatric disorder), personal history of COVID-19 infection, fear of COVID-19-induced death, distrust toward the government, the thought of embarrassment of getting infected with COVID-19, and presence of severe psychological distress.

## 4. Discussion

The proportion of COVID-19 vaccine hesitancy among the Japanese population in this study was 11.3%. The proportion was higher among younger respondents and female respondents. Among respondents who did not want to get vaccinated against COVID-19, the top reason given was concern about adverse reactions, which was mentioned by more than 70% of the Hesitant group, followed by doubts about the efficacy of the vaccine, which was mentioned by 20%. Factors associated with the hesitancy were female sex, living alone, lower socioeconomic status, and presence of severe psychological distress, which were more apparent among older respondents than among younger respondents.

Our results of the proportion of the hesitancy among the Japanese population are comparable to that of other countries. However, they differ from the survey conducted in January 2021 (*N* = 3000) [6], which reported a hesitancy of 37.9%, but this may be due to the definition of the hesitancy and whether or not demographic background was adjusted. In the previous study, the answer “unsure” was included in the definition of the hesitancy. In addition, although the study accounted for both sex and age during sampling, they did not make any adjustment for demographic background when the proportion was determined, which may have led to the discrepancy in their results. In fact, in that study, almost half of the respondents were distributed in the lowest income category. In the present study, we used a nationally representative sample (CSLCPHW) to make the adjustment, and our analysis is as close as possible to the distribution of the general population in terms of demographic background.

The most commonly cited reason for not getting vaccinated was concern about adverse reactions in more than 70% of the respondents. In addition, distrust toward the government and toward government policy on COVID-19 were observed to be factors related to the hesitancy, which is consistent with previous studies [23,24,25]. Other cited reasons for the hesitancy unique to the COVID-19 vaccine were the new mechanism of administration of mRNA by some vaccines, and the fast vaccination approval process [14]. In a report released by the WHO, consistent, transparent, empathetic, and proactive communication about vaccines was cited to help build trust in COVID-19 vaccines [26]. For example, the use of trusted messengers and greater transparency in the vaccine development process will help to increase confidence in COVID-19 vaccines.

Interestingly, the proportion of COVID-19 vaccine hesitancy in older respondents was 4.8% among male respondents and 7.7% among female respondents, and that in younger respondents was 14.2% and 15.6%, respectively; the difference between the older and younger respondents was more than double. The reasons for not getting vaccinated were the same in both older and younger respondents in terms of adverse reactions and insufficient efficacy; however, younger respondents differed in terms of lack of time and belief that they would not become seriously ill. The reasons for getting vaccinated were the same in older and younger respondents in terms of concerns about becoming infected themselves or passing the disease on to family members or others; however, older respondents were more likely than younger respondents to cite risk of serious illness and the need for vaccination to benefit society. These results suggest the necessity of using different methods for younger and older people when making vaccination recommendations.

Furthermore, the risk factors for vaccine hesitancy were female sex, living alone, low socioeconomic status, and the presence of psychological distress, which were more apparent in older respondents than in younger respondents. Recent research suggests that people with lower socioeconomic status and severe psychological distress are more vulnerable to susceptibility to disinformation and misinformation about COVID-19 [27,28]. While all people should be vaccinated, older individuals are particularly at high risk of severe infection with COVID-19 [29], and priority should be given to this population to reduce preventable deaths. Targeted strategies to raise awareness that negative feelings about vaccines can be manipulated through disinformation campaigns, or to elicit positive feelings that vaccination contributes to the health and well-being of the community, could be useful in activating COVID-19 vaccine confidence [30].

Although the strength of this study was its large-scale sample (*N* = 26,000) covering all prefectures in Japan, there are some limitations. First, the data were collected by an Internet survey. However, possible selection bias (differences in sociodemographic status between the Internet survey participants in the present study and the general Japanese population) was accounted for by making adjustments as much as possible using an external, nationally representative sample (see the Appendix A). Second, the cross-sectional design meant that we could not draw any conclusions on the direction of causality. However, our study aim was not to clarify causal direction, but rather to identify the proportion of the vaccine hesitancy and its related factors. Therefore, the cross-sectional design was not considered to have any adverse effect on the relevancy of our results. Lastly, we did not assess factors associated with fluctuations in vaccine hesitancy, such as COVID-19 misinformation/infodemic, the influence of politicians’ actions, and observed side effects such as abnormal blood clotting. Future research needs to focus on the patterns of hesitancy over time and the key phenomena that influenced sharp decreases and increases in hesitancy.

## 5. Conclusions

The proportion of COVID-19 vaccine hesitancy in Japan was comparable to that in previous studies conducted in other countries, and the proportion among younger respondents was more than double that of older respondents. Concerns about adverse reactions as a reason for their not taking the vaccine was cited by more than 70% of the respondents. Factors associated with COVID-19 vaccine hesitancy were female sex, living alone, low socioeconomic status, and presence of severe psychological distress, especially among older respondents. Thus, adequate measures should be taken to ensure that vaccines are delivered to people with these factors.

## Figures and Tables

**Table 1 vaccines-09-00662-t001:** Participants’ baseline characteristics included in analyses.

	Total (*N* = 23,142)		COVID-19 Vaccine Intention	
			Intend (*N* = 20,527)	Hesitant (*N* = 2615)
	*N*	%	*N* (%)	*N* (%)
***Sex***, Female	11,376	49.2	9955 (87.5)	1421 (12.5)
***Age group (years)***				
Younger (15–39)	6686	28.9	5690 (85.1)	996 (14.9)
Middle-aged (40–64)	10,554	45.6	9304 (88.2)	1250 (11.8)
Older (65–79)	5903	25.5	5533 (93.7)	369 (6.3)
***Income (million JPY/year)***				
less than 1	841	3.6	645 (76.7)	196 (23.3)
1 to less than 6	10,699	46.2	9496 (88.8)	1203 (11.2)
6 to less than 120	5433	23.5	4953 (91.2)	480 (8.8)
120 or more	952	4.1	863 (90.7)	88 (9.3)
No response/unknown	5217	22.5	4569 (87.6))	648 (12.4)
***Marital status***				
Never married	5697	24.6	4769 (83.7)	928 (16.3)
Married	15,262	65.9	13,824 (90.6)	1437 (9.4)
Widowed/divorced	2183	9.4	1934 (88.6)	249 (11.4)
Living alone, yes	3076	13.3	2618 (85.1)	458 (14.9)
***Occupation***				
Health care professional	877	3.8	789 (89.9)	88 (10.1)
Essential worker in the food industry	3381	14.6	3010 (89.0)	371 (11.0)
Other occupation	9878	42.7	8643 (87.5)	1235 (12.5)
Unemployed	9006	38.9	8086 (89.8)	920 (10.2)
***Educational level***				
Junior high school graduate	4114	17.8	3597 (87.4)	517 (12.6)
High school graduate	9739	42.1	8654 (88.9)	1085 (11.1)
Two-year college graduate	3366	14.5	2903 (86.3)	462 (13.7)
Bachelor’s degree	4315	18.6	3876 (89.8)	439 (10.2)
Master’s or doctoral degree	1608	7.0	1497 (93.1)	112 (6.9)
***Use of combustible cigarettes or HTPs***				
Neither	19,024	82.2	16,833 (88.5)	2190 (11.5)
Only HTPs	2286	9.9	2055 (89.9)	231 (10.1)
Only combustible cigarettes	915	4.0	796 (87.0)	119 (13.0)
Dual use	917	4.0	842 (91.8)	75 (8.2)
***Alcohol use***				
Never	7269	31.4	6159 (84.7)	1110 (15.3)
Ever	7717	33.3	6929 (89.8)	788 (10.2)
Current	8156	35.2	7439 (91.2)	717 (8.8)
***Comorbidity (present)***				
Hypertension	4490	19.4	4146 (92.3)	344 (7.7)
Diabetes mellitus	1628	7.0	1531 (94.0)	97 (6.0)
Asthma or COPD	1031	4.5	922 (89.4)	109 (10.6)
Cardiovascular disease	310	1.3	269 (86.6)	41 (13.4)
Cerebrovascular disease	260	1.1	232 (89.2)	28 (10.8)
Cancer	442	1.9	398 (90.1)	43.9 (9.9)
Chronic pain	2330	10.1	2089 (89.7)	241 (10.3)
Psychiatric disorder	1463	6.3	1314 (89.8)	149 (10.2)
Personal history of COVID-19 infection	400	1.7	331 (82.8)	69 (17.2)
Fear of COVID-19-induced death	7884	34.1	7285 (92.4)	599 (7.6)
Perceived likelihood of getting infected with COVID-19 themselves	3197	13.8	2880 (90.1)	317 (9.9)
Distrust toward the government	16,962	73.3	14882 (87.7)	2079 (12.3)
Distrust toward government policy on COVID-19	15,724	67.9	13787 (87.7)	1937 (12.3)
Thought of embarrassment of getting infected with COVID-19 themselves	5463	23.6	4841 (88.6)	622 (11.4)
Severe psychological distress, yes	2741	11.8	2299 (83.9)	442 (16.1)
Living in a prefecture with a high proportion of COVID-19 cases	7729	33.4	6863 (88.8)	866 (11.2)

Analyses were weighted to adjust for differences between Internet survey respondents in the present study and nationally representative samples. The weighting meant that the sum of participants did not necessarily equal the total number of participants. All numbers are percentages (%), except those for N. Abbreviations: JPY, Japanese yen; HTP, heated tobacco product; COPD, chronic obstructive pulmonary disease; COVID-19, coronavirus disease 2019.

**Table 2 vaccines-09-00662-t002:** Proportion of COVID-19 vaccine hesitancy, stratified by age and sex.

	COVID-19 Vaccine Intention	
	No. of Hesitant/Total	% (95% CI)
All	2615/23142	11.3 (10.9–11.7)
Male respondents	1194/11766	10.2 (9.6–10.7)
Younger (15–39)	465/3277	14.2 (13.0–15.4)
Middle-aged (40–64)	586/5514	10.6 (9.8–11.5)
Older (65–79)	143/2975	4.8 (4.1–5.6)
Female respondents	1421/11376	12.5 (11.9–13.1)
Younger (15–39)	531/3408	15.6 (14.4–16.8)
Middle-aged (40–64)	664/5040	13.2 (12.3–14.1)
Older (65–79)	226/2928	7.7 (6.8–8.7)

Analyses were weighted to adjust for differences between Internet survey respondents in the present study and nationally representative samples. The weighting meant that the sum of participants did not necessarily equal the total number of participants. Abbreviations: COVID-19, coronavirus disease 2019; CI, confidence interval.

**Table 3 vaccines-09-00662-t003:** Reasons for getting vaccinated and not getting vaccinated against COVID-19.

	COVID-19 Vaccine Intention	
	Intend, *N* (%)	Hesitant, *N* (%)
***Reasons for getting vaccinated, %***		
It was recommended by a family member or friend	568 (2.8)	-
It was recommended by SNS or the media	397 (1.9)	-
I’m worried about getting infected with COVID-19	6807 (33.2)	-
I think I have a high risk of becoming seriously ill	3516 (17.1)	-
I am a medical worker	766 (3.7)	-
I don’t want to infect my family or other people around me.	6853 (33.4)	-
I think it is necessary for society to be vaccinated	6510 (31.7)	-
I can get it for free	3795 (18.5)	-
***Reasons for not getting vaccinated, %***		
I don’t have time to go get vaccinated	-	229 (8.8)
I’m worried about adverse reactions	-	1934 (73.9)
I don’t think it is very effective	-	508 (19.4)
I don’t think I will get infected	-	202 (7.7)
I think I have a low risk of getting seriously ill	-	197 (7.5)
I was previously infected with COVID-19	-	12 (0.5)
I have already received the COVID-19 vaccine	-	10 (0.4)
It was recommended by a family member or friend	-	34 (1.3)
It was recommended by SNS or the media	-	72 (2.8)

Analyses were weighted to adjust for differences between Internet survey respondents in the present study and nationally representative samples. The weighting meant that the sum of participants did not necessarily equal the total number of participants. Abbreviations: COVID-19, coronavirus disease 2019; SNS, social network services.

**Table 4 vaccines-09-00662-t004:** Multivariate logistic regression models for odds ratio of COVID-19 vaccine hesitancy.

	aOR	95% CI	*p*-Value
***Sex***, female	1.20	(1.09–1.32)	<0.01
***Age group (years)***			
Younger (15–39)	1.04	(0.94–1.16)	0.46
Middle-aged (40–64)	ref	-	-
Older (65–79)	0.58	(0.50–0.67)	<0.0001
***Income (million JPY/year)***			
less than 1	1.78	(1.49–2.14)	<0.0001
1 to less than 6	ref	-	-
6 to less than 120	0.74	(0.66–0.83)	<0.0001
120 or more	0.79	(0.62–0.99)	0.04
No response/unknown	0.98	(0.88–1.09)	0.75
***Marital status***			
Never married	ref	-	-
Married	0.80	(0.71–0.89)	<0.0001
Widowed/divorced	0.81	(0.68–0.96)	0.01
Living alone, yes	1.29	(1.12–1.47)	<0.01
***Occupation***			
Health care professional	0.73	(0.58–0.93)	0.01
Essential worker in the food industry	0.85	(0.74–0.96)	0.01
Other occupation	ref	-	-
Unemployed	0.85	(0.77–0.94)	<0.01
***Educational level***			
Junior high school graduate	1.19	(1.04–1.38)	0.01
High school graduate	1.10	(0.98–1.25)	0.11
Two-year college graduate	1.29	(1.11–1.49)	<0.01
Bachelor’s degree	ref	-	-
Master’s or doctoral degree	0.91	(0.71–1.15)	0.41
***Use of combustible cigarettes or HTPs***			
Neither	ref	-	-
Only HTPs	0.95	(0.82–1.10)	0.48
Only combustible cigarettes	1.24	(1.01–1.53)	0.04
Dual use	0.70	(0.55–0.90)	0.01
***Alcohol use***			
Never	ref	-	-
Ever	0.66	(0.59–0.73)	<0.0001
Current	0.63	(0.57–0.70)	<0.0001
***Comorbidity (present)***			
Hypertension	0.88	(0.77–1.00)	0.05
Diabetes mellitus	0.58	(0.46–0.73)	<0.0001
Asthma or COPD	1.06	(0.85–1.33)	0.61
Cardiovascular disease	1.74	(1.18–2.57)	0.01
Cerebrovascular disease	1.05	(0.68–1.62)	0.84
Cancer	1.01	(0.71–1.43)	0.95
Chronic pain	1.01	(0.87–1.17)	0.93
Psychiatric disorder	0.64	(0.53–0.77)	<0.0001
Personal history of COVID-19 infection	1.30	(0.98–1.74)	0.07
Fear of COVID-19 induced death	0.54	(0.49–0.60)	<0.0001
Perceived likelihood of getting infected with COVID-19 themselves	1.03	(0.90–1.18)	0.70
Distrust toward the government	1.28	(1.11–1.46)	<0.01
Distrust toward government policy on COVID-19	1.24	(1.09–1.41)	<0.01
The thought of embarrassment of getting infected with COVID-19	1.06	(0.96–1.17)	0.29
Severe psychological distress, yes	1.43	(1.26–1.61)	<0.0001
Living in a prefecture with a high proportion of COVID-19 cases	1.02	(0.93–1.12)	0.68

Analyses were weighted to adjust for differences between Internet survey respondents in the present study and nationally representative samples. Abbreviations: JPY, Japanese yen; HTP, heated tobacco product; COPD, chronic obstructive pulmonary disease; COVID-19, coronavirus disease 2019; aOR, adjusted odds ratios; CI, confidence interval.

## Data Availability

All anonymized individual participant data reported in this paper are available for interested researchers who send a request for data sharing, along with a synopsis of the secondary analysis plan paper to the corresponding author (R.O.).

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
