# Peer review of "COVID-19 Vaccine Hesitancy and Its Associated Factors in Japan"

_vaccines, 2021, doi:10.3390/vaccines9060662_

Round 1
Reviewer 1 Report
The authors have conducted a nationwide, cross-sectional internet survey among N=23,142 respondents to clarify the proportion of vaccine hesitancy in Japanese general population. They main reason reported for not getting vaccinated was concerns about adverse reactions in more than 70% of the respondents. In addition, the factors associated with vaccine hesitancy was examined. Younger respondents were more hesitant for getting vaccinated when compared to older respondents. Further the authors have generalized the results using the 2016 Comprehensive survey of Living Conditions of People on Health and Welfare (CSLCPHW). Finally, this cross-sectional survey has added supporting evidence into the knowledge about vaccine hesitancy in Japan and warranted for adequate measures to ensure the vaccines are delivered to all people including vulnerable populations.
I have few comments that needs to be addressed.
- Page 1, Introduction: Update the Covid -19 prevalence during the revision stage. The Covid-19 infection rate has increased and would be good to have the updated numbers in the revision stage before publication.
- Page 5, Line 171-173. From Table 3, I understand that the reason for not getting vaccinated was concern about adverse reactions in 70% of the participants. The exact numbers from the Table 3 was 73.9%. Either say more than 70% or use correct percentage (73.9%) in the main manuscript text. In abstract, you have mentioned “more than 70%” and that is fine.
- Page 5, Line 172-173. Similar as above, From Table 3, the doubts about the efficacy of the vaccine was 30% as mentioned in the manuscript text. But from Table 3, the “I don’t think it is very effective” response was 19.4%. Could you clarify this percentage difference and mention the correct percentage?
- Page 6, Line 188. From Table 4, the factors associated with Covid-19 vaccine hesitancy were “never married” as mentioned in the manuscript text. However, from Table 3, the never married was used as a reference category. So, you must modify as the factors associated with Covid-19 vaccine hesitancy were married, living alone in the manuscript text.
- Page 6, Line 190: Mention all the factors associated with Covid-19 vaccine hesitancy from Table 4 including the current alcohol use, presence of comorbidities (diabetes mellitus, psychiatric disorder). You have mentioned only very few factors in the main manuscript text. It is a good to mention all the factors associated with Covid-19 vaccine hesitancy and it common to see people with comorbidities are vaccine hesitant.
- Page 6, Line 193: Table S5: Present the main results of the supplementary Table S5 in the main manuscript text. Describe only main results which are consistent with the stratified age group and this will facilitate the reader about the risk factors in overall respondents (Table 4) as well as the how the perception differed with respect to different age groups (Table S5).
- Page 8, Line 205-206: Rephrase the sentence as “Reasons for not getting vaccinated against COVID-19 were concerns about adverse reactions in more than 70%, followed by doubts about the efficacy of the vaccine in 20% of the respondents”. The current text in the manuscript “Reasons for not getting vaccinated against COVID-19 were concerns about adverse reactions, followed by doubts about the efficacy of the vaccine in 70% of the respondents” is misleading on reporting adverse reactions and efficacy of the vaccine.
- Page 8, Line 223, Rephrase as “adverse reactions in more than 70% of the respondents”.
- Supplementary File, Table S4, Add a space between “Reasons for getting vaccinated”. Now the word “forgetting” is in the title which is incorrect. This is not on page 1, but on page 3 Table 4 title.
Reviewer 2 Report
Title: COVID-19 vaccine hesitancy and its associated factors in Japan
Manuscript-ID: vaccines-1212750
The paper deals with the important issue of COVID-19 vaccine hesitancy and factors affecting it in the Japan population. The study found an overall vaccine hesitancy of 11.3% (10.9-11.7%) in the Japan population. While vaccine hesitancy was lowest among younger female respondents (15.6%), the most popular reasons for not intending to get vaccinated were concern about vaccine adverse reactions and doubts on the efficacy of the vaccines. The large sample size of the study with a national coverage is a main strength of the paper. However, there some important issues that the authors may want to consider in their revision (detailed below).
Introduction:
The introduction needs a revision:
The introduction states “To date, two studies have examined vaccine hesitancy in Japan, but the re-49 ported proportions of the hesitancy have been inconsistent.” There have been huge fluctuations in COVID-19 vaccine hesitancy almost throughout the world and there were important reasons for this such as the COVID infodemic, reports of blood clotting side effects from some of the vaccine types and so on. As a result, it is not expected that studies conducted in September and January in would have similar findings. Therefore, it is better to state those previous findings and justify the need for current research in terms of checking the rapidly changing hesitancy patterns instead of inconsistency in the findings of the previous studies. In addition, I wouldn’t use “clarify” as in “…clarify the proportion of COVID-19 vaccine hesitancy”.
Again in the introduction, the following paragraph in confusing and needs revision:
“Furthermore, studies have consistently reported that COVID-19 vaccine hesitancy is possibly associated with female sex, young age, low income, and low education level [3]. However, both of the abovementioned studies in Japan examined factors that influence willingness to get vaccinated against COVID-19, whereas risk factors for vaccine hesitancy have remained unclear.” What is the difference between “factors” and “risk factors” in your paragraph? The message from this paragraph is that “factors” have been dealt in the previous studies but not “risk factors” and therefore this study deals with the later. Do you mean that? What are the things you are considering as “risk factors”? But then in the next paragraph you stated “ Therefore, in this study, we aimed to clarify the proportion of COVID-19 vaccine hesitancy among the Japanese population using a larger sample than employed in previous studies, and to identify factors associated with COVID-19 vaccine hesitancy…”
The last sentence of the introduction should go to the methods section.
Materials and Methods:
Page 2 line 73: “Briefly, we conducted a baseline 73 survey during the summer of 2020…” For purposes of clarity, it is better to mention the months instead of “summer” which is different in different parts of the world.
Page 2 line 78: Can you please give a brief explanation of what “prefecture” is in a parenthesis or footnote at its first use?
Page 2 line 81: “After the baseline survey, we conducted a follow-up survey from February 8 to 26, 2021. Of the 28,000 participants in the baseline survey, 24,059 participated in the follow-up survey. In addition, we recruited 1941 new general residents aged 15-79 years, for a total of 26,000 study participants.” Does this mean hesitancy was measured twice? If yes, this can be another major strength of the study and can be highlighted in the introduction.
Page 2 line 83: “In addition, we recruited 1941 new general residents aged 15-79 years, for a total of 26,000 study participants.” Needs a revision for clarity.
Page 2 line 90: “As an incentive for study 90 participation, the participants received credit points called “Epoints,” which can be used 91 for online shopping and cash conversion.” It would be good to mention the amount of money offered.
Page 3 line 105: I assume that the expression “Demographic factors found to be possibly related to COVID-19 vaccine hesitancy 105 were as follows…” is written as a justification for the included independent variables and is based on findings of previous studies. If I am right, two things need to addressed; 1) The fact that this is being presented as a justification for the factors included and taken from previous studies should be clearly stated, 2) after listing a massive number of factors, only one reference is cited. Factor selection should ideally consider as many as possible of previous study findings giving priority to studies of high-level evidence such as systematic reviews and meta-analyses.
Results
- Page 4 line 154: I am not sure what you mean by “From among the 26,000 total participants, 2,858 who provided straight lining or discrepant responses were excluded.” If you mean those people who switched from “Intend” to “Hesitant” or vice versa between the two surveys were excluded, I think that is a loss of an important information. Instead, a good analysis and description of the switching and associated factors is very important.
- Was there a reason why the authors preferred to report only percentages and no numbers in all the tables? I think it is much easier for readers to read a table having numbers than only percent. Therefore, I would revise all table cells to include numbers as n (%). It is also important to include the numbers of people in each categories such as “Intend” and “Hesitant” in all the tables.
- Simple age range categorisation is used in Table 1 while the terms younger, middle-aged and older are used in other tables. 1) There should be consistency in presentation throughout the paper, 2) if the authors prefer to include the nominal terms younger, middle-aged and older categorisations, it is necessary to have reference/s for this.
Discussion
- In my opinion, a major weakness of the paper is the lack of attention given to important broader factors that were responsible for hesitancy floatation everywhere. These include the COVID misinformation/infodemic, influence of politicians’ actions, observed side effects such as clotting. Although the authors didn’t have the opportunity to include these factors in their study, it is important to have a look into the patterns of hesitancy over time and important phenomenon that affected the sharp drop or increase in hesitancy. Mentioning only the factors such inclusion of “unsure” category as explanation of differences in the findings ignores the higher-level factors.
- A lot of the discussion is not sounds like a result section in that it is full of the numeric findings. A discussion section should highlight only key findings without repeating the results section and most of the discussion should be reserved for things such comparison with previous studies, explaining discrepancies, the research and policy implications of the findings, limitations of the study, etc.
- Page 9 line 249: Personally, I find the following information inappropriate and misleading: “Because older individuals are at higher risk of severe COVID-19 infection [23], this population needs to be vaccinated to reduce preventable deaths.” Ideally, everyone, unless there are medical reasons, should be vaccinated and that is what researchers should encourage. Older people are usually at greater risk of death and therefore it is more important these group of people are given priority.
- Page 9 line 254: “First, the data were collected by an Internet survey. However, possible bias was accounted for by making adjustments as much as possible using an external, nationally representative sample.” This is not clear to me. What kind of bias are you talking about.
- Page 9 line 264: “Concerns about adverse reactions as a reason for their not taking the vaccine was cited by more than 70% of the respondents. Factors associated with COVID-19 vaccine hesitancy were female sex, living alone, low socioeconomic status, and presence of severe psychological distress, especially among older respondents. Thus, so adequate measures should be taken to ensure that vaccines are delivered to people with these factors.” I think you are saying people with these factors are hesitant to get vaccinated. So what do you mean by “so adequate measures should be taken to ensure that vaccines are delivered to people with these factors”.
Other minor comments:
The authors need to consider a good language editing as there are several grammar and other language issues throughout the paper. The following are some examples:
Page 2 line 71: This was a cross-sectional study
Page 2 line 100: The survey also asked both the “Intend” and “Hesitant” groups about reasons for intending to get vaccinated and not to do so, respectively.
Page 3 line 125: Living in an area with a higher proportion of people infected with COVID-19 virus was calculated.
Page 3 line 134: We adjusted for differences between the Internet survey respondents and the general 134 population (e.g., younger individuals are more likely to participate in and respond
Page 4 line 171: “As shown in Table 3, the reasons for not wanting to get vaccinated were concern about adverse reactions in 70% of the participants, followed by doubts about the efficacy of the vaccine in 30%.”
